# Animal–Energy Relationships in a Changing Ocean: The Case of Continental Shelf Macrobenthic Communities on the Weddell Sea and the Vicinity of the Antarctic Peninsula

**DOI:** 10.3390/biology12050659

**Published:** 2023-04-27

**Authors:** Enrique Isla

**Affiliations:** Institut de Ciències del Mar-CSIC, 08003 Barcelona, Spain; isla@icm.csic.es

**Keywords:** climate change, polar warming, biological response, ecosystems, biodiversity conservation, vulnerability

## Abstract

**Simple Summary:**

Antarctic macrobenthic communities thriving on the continental shelf have evolved over thousands of years in perhaps the coldest and most constant environmental conditions in the ocean. This situation may have limited their genetic plasticity to cope with relatively fast environmental changes. The Antarctic geophysical setting restricts the generation of the organic energy (e.g., organic carbon) that fuels continental shelf macrobenthos to the spring and summer months, when vigorous primary production develops in the water column. Wind, ice, and water currents modulate pelagic energy production and distribution over the shelf in such coupled harmony, at least since the last glacial maximum, that it enables the existence of abundant and diverse macrobenthic communities over thousands of square kilometers. Ongoing environmental change, especially warming, is expected to modify the ancestral balance among energy and macrobenthic communities’ abundance on the Antarctic continental shelf. Most likely, the current macrobenthic community assemblages will change, presumably first affecting the east of the Weddell Sea over the rest of the region. The demise of the rich biodiversity of Antarctic macrobenthos will impede our knowledge of Antarctic ecology and our ability to reach a complete understanding of several aspects (e.g., evolutionary, physiological, trophic, and reproductive) of these old marine animals.

**Abstract:**

The continental shelves of the Weddell Sea and the Antarctic Peninsula vicinity host abundant macrobenthic communities, and the persistence of which is facing serious global change threats. The current relationship among pelagic energy production, its distribution over the shelf, and macrobenthic consumption is a “clockwork” mechanism that has evolved over thousands of years. Together with biological processes such as production, consumption, reproduction, and competence, it also involves ice (e.g., sea ice, ice shelves, and icebergs), wind, and water currents, among the most important physical controls. This bio-physical machinery undergoes environmental changes that most likely will compromise the persistence of the valuable biodiversity pool that Antarctic macrobenthic communities host. Scientific evidence shows that ongoing environmental change leads to primary production increases and also suggests that, in contrast, macrobenthic biomass and the organic carbon concentration in the sediment may decrease. Warming and acidification may affect the existence of the current Weddell Sea and Antarctic Peninsula shelf macrobenthic communities earlier than other global change agents. Species with the ability to cope with warmer water may have a greater chance of persisting together with allochthonous colonizers. The Antarctic macrobenthos biodiversity pool is a valuable ecosystem service that is under serious threat, and establishing marine protected areas may not be sufficient to preserve it.

## 1. Introduction

The Southern Ocean’s (SO) great spatial extent makes it especially sensitive to global change effects [1]. For example, the SO has stored more than 40% of the global upper 2000 m ocean heat gain from 1970 to 2017. Nevertheless, during the period from 2005 to 2017, these proportions were even higher, reaching values between 45% and 62% and causing a significant warming of the basin [2]. The ongoing intensification of westerly winds due to atmospheric warming over the SO increases heat transport toward the south, enhancing ice shelf melting and reducing sea ice extent [3,4]. In this ongoing atmospheric and ocean-warming scenario, projections suggest that primary production will increase in the SO [5], as already occurs at the western coast off the Antarctic Peninsula, where primary production has developed in recent sea ice-free areas [6,7]. The ongoing and projected increases in physical (e.g., wind and heat) and chemical (e.g., primary-produced organic carbon) energy at the sea surface should produce an impact on the seabed, where abundant benthic life benefits from the export of pelagic energy to the deep. However, little is known about this potential impact, and a general picture of the ongoing sea surface and seabed environmental changes and their connections is still lacking. Based on a literature review and my own experience, I attempt to develop a general picture of animal–energy relationships at the Weddell Sea and the western Antarctic Peninsula continental shelves, bringing together several physical, chemical, geological, and biological aspects of the local environments and the macrobenthic fauna thriving there.

## 2. Objectives

To provide a synthetic picture of the ongoing animal–energy relationships at the Weddell Sea and the western Antarctic Peninsula continental shelves and their potential consequences determined by the ongoing global change pressure on an apparently remote region of the ocean.

## 3. Methods

The synthetic information presented in this analysis follows a path from the sea surface to the seafloor, using organic carbon budgets to structure the energy flow from primary production to macrobenthic biomass stocks, modulated by the current environmental controls at the sea surface and along the water column.

## 4. Results

### 4.1. General Aspects of the Physical Environment of the Weddell Sea and the Vicinity of the Antarctic Peninsula Continental Shelves

The Antarctic continental shelf has extraordinary characteristics relative to the global ocean. It is not physically connected to any other continental shelf in the ocean, and it is also surrounded by the Antarctic Circumpolar Current, the largest water current in the ocean, which also functions as a relatively cold geophysical barrier. The Antarctic continental shelf is on average 500 m deep, which is approximately eight times the world average, and in the western Weddell Sea (WS), areas larger than 450 km wide also make the width average higher than anywhere else [8]. Floating ice shelves permanently cover more than one third of the Antarctic continental shelf [9], and in places such as the eastern WS, the Riiser-Larsen ice shelf even reaches the continental shelf break. Along the WS, this bathymetric point is typically shallower than the inner shelf, showing a cross-shelf profile sloping toward the south, where in the central WS, the axis of the Filchner Trough can reach more than a 1000 m water depth (Figure 1a) [8].

During the autumn and winter, sea ice completely covers the continental shelf, drastically reducing light availability and pelagic primary production in the water column. These seasonal conditions remain for approximately 6 months over most of the WS continental shelf; however, to the west, multiyear sea ice develops perennial ice cover on the continental shelf off the eastern coast of the Antarctic Peninsula (AP). This sea ice cover pattern creates a longitudinal gradient along the WS continental shelf from seasonal open-water and productive conditions to the east to an irregular ice-covered and intermittent primary productive environment to the west [10]. 

Ice also controls sedimentation on the WS; the absence of rivers and the presence of ice shelves restrict sediment inputs mainly to lateral transport along the continental shelf, glacial marine sources from the grounding line of ice shelves, and ice-rafted debris [11,12]. Consequently, lithogenic sediment inputs are comparatively low relative to lower latitude settings. Sediment accumulation rate studies are scarce and limited to the eastern half of the WS, where the results vary between 1 cm ky^−1^ and 5 cm ky^−1^ [13], and the northwestern WS, where accumulations are found between 140 cm ky^−1^ and 250 cm ky^−1^ [14]. Sediment accumulation rate differences are most likely related to the diversity in the geophysical settings, where insular and peninsular inputs enhanced by rainfall and higher temperatures at the northwestern WS increase sediment accumulation rates by orders of magnitude relative to the eastern section.

Wind is another physical agent that enhances sedimentation on the continental shelf. On the one hand, the wind force can break sea ice cover at the sea surface, creating highly productive open-water latent heat polynyas [15,16], where sinking biogenic particle fluxes originate [17,18]. Polynyas can also develop due to the upwelling of warm deep water (sensible heat polynya), where high primary production rates also take place, stimulated by iron fertilization coming from melting ice shelves [19,20]. On the other hand, over open water or melting sea ice conditions, wind can stimulate primary production and particle aggregation, producing sediment pulses larger than 600 mg m^−2^ d^−1^, which account for tons of biogenic and lithogenic materials settling over large portions of the shelf (tens of km^2^) just in a few days [21].

Near the seabed, water currents mainly follow a diurnal, spring-neap tidal pattern [22]. The deep continental shelf contributes making near-bottom water current circulation sluggish, although the top velocities may reach values above 30 cm s^−1^ [8,22]. However, near the continental shelf break, where deep currents intrude on the shelf, velocities are faster and can reach values above 50 cm s^−1^ [22]. Water coming onto the continental shelf from the slope transports heat from the Antarctic circumpolar deep water, exposing the benthic environment to periodical warming events reaching even 2 °C [22]. Warm water intrusions disrupt the usual below 0 °C conditions, severely compromising the viability of the local stenothermal communities, which have evolved over thousands of years during a relatively constant temperature, resulting in low genetic plasticity (mainly due to the long generation time observed for many of them) to cope with heat stress [23]. Iceberg scouring also modifies the benthic environment, even at a 600 m water depth, by brutally disturbing the upper layers (several meters) of the seabed. This process completely removes the benthic macrofauna from the sea floor, developing a patchy disturbance (and distribution) pattern, which may occur over the Antarctic continental shelf once every 200 y m^−2^ [24,25].

The WS limits to the west with the AP, on average, a 1500 m-high topographical obstruction that blocks the westerly winds blowing into the WS, developing a natural east to west climatic gradient at the sea surface, from the warmer, humid, and open-water environment off the northern coast of the South Shetland Archipelago at the Drake Passage (DP) over a transitional region across the Bransfield Strait (BS), to a cold, dry, and polar setting at the northwestern WS [26,27,28]. Sea ice extent and chlorophyll-a abundance vary from low to high values along the same geographical direction [29]. Contrastingly, the water temperature gradient near the sea floor goes from values above 0 °C over the continental shelf off the South Shetland Islands (SSI) to freezing temperatures on the shelves of the northwestern WS and the BS [30,31].

Unlike the WS, the continental shelves in the BS and off the SSI are narrow (5 km to <60 km wide) and incised by a number of troughs, with such a shape and interspace (~15 km in the BS and ~25 km in the DP) that make the continental shelf geomorphology in this region unique in the Antarctic [29]. Sediment supply to the Strait is largely related to the materials coming from the AP and the SSI, mainly consisting of diatomaceous mud and ice-rafted debris, in addition to the local pelagic biogenic (e.g., organic matter and biogenic silica) inputs [14,32]. The set of troughs across the continental shelf act as sediment traps that block the along-shelf sediment transport near the sea floor, restricting sedimentation rates. Nevertheless, sediment accumulation rates in this area are larger relative to the eastern WS continental shelf, varying from 60 cm ky^−1^ to 270 cm ky^−1^ [14,33]. Water circulation, lateral transport, and sea floor relief make sedimentation processes complex and diverse within the region [14]. Water circulation over the continental shelf off the western AP and off the northern coast of the SSI is ~7 cm s^−1^; although, on the peninsular side, currents may reach 50 cm s^−1^ during winter, enhancing particle transport toward the southwest [34,35,36]. At the sea surface, water circulation in combination with the rugged coastline of the western AP and the SSI enhance the stratification of the water column and water residence time within the coastal fjords, producing a biological environment of high production that converts the continental shelf of the region in a carbon sink. The continental shelf sediment column in this area (e.g., Bransfield and Gerlache Straits) can store up to 5% of the annual primary production of organic carbon, in contrast to the 1% to 2% of the global ocean’s average [33].

### 4.2. Macrobenthos and Biological Characteristics of the Environment

When the seasonal sea ice extent diminishes, pelagic primary production takes place over the WS continental shelf, developing mainly from November to March. It peaks in December, when it can reach more than 1.4 g C m^−2^ d^−1^, although values > 1 g C m^−2^ d^−1^ remain until February; these numbers add for an annual primary production over the WS shelf of approximately 17 Tg C y^−1^ [37]. During the productive season, the wind stimulates biogenic matter pulses that deliver hundreds of tons of organic carbon (OC) and biogenic silica (bSi) over the continental shelf in just a few days [21]. As a consequence, green mats (also described as “food banks” for the AP fjords) of fresh labile materials (e.g., lipids) blanket hundreds of square kilometers of the seabed [38,39]. OC found in the WS continental shelf sediment represent up to 97% (weight) of the total sediment C and showed comparatively high lipid (~3 mg g^−1^) and protein (~5 mg g^−1^) contents within a global context, which were even higher than in other Antarctic settings, such as the Ross Sea [40]. It has been found that lipids are the best descriptors for the most labile OC fractions in the sediment column, where they have also showed significant associations with benthic meiofauna abundance and biomass [41,42]. However, in spite of the presence of such amounts of organic energy settled on the seabed, the OC content in the WS continental shelf sediment is not especially high, showing values between 0.2% and 1.2% (weight) (Figure 1b) [40,43]. The OC contents in the sediment of the narrower shelf off the AP coast, including the Bransfield and Gerlache Straits, also showed values between 0.2% and 1.1% (weight) (Figure 1b) [14,33], even under areas with comparatively high primary production rates, such as the Gerlache Strait, where values can reach 5 g C m^−2^ d^−1^ [33]. Hence, the sediment OC contents in all these regions point to efficient benthic consumption and lateral transport.

**Figure 1 biology-12-00659-f001:**
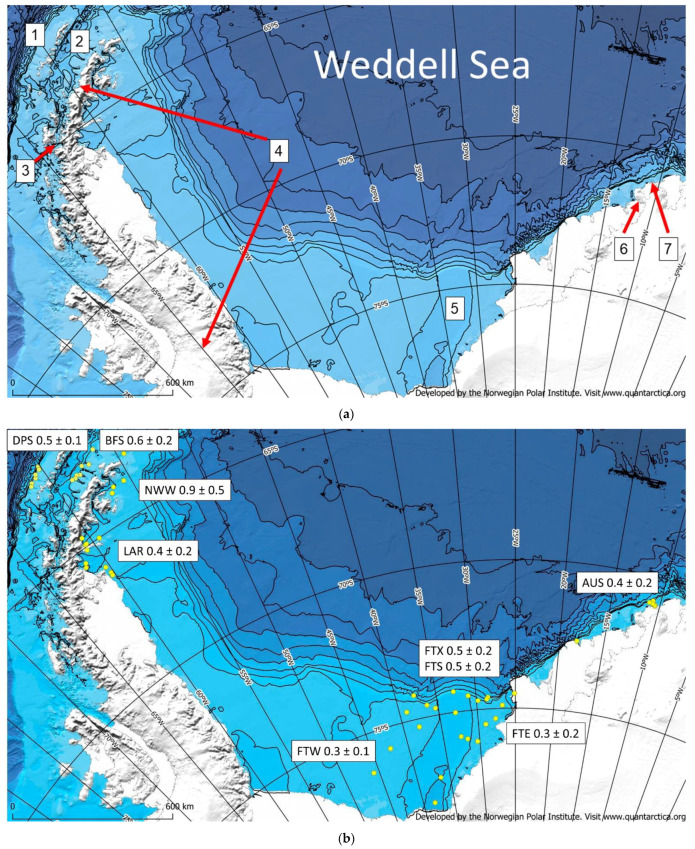
(**a**) Study area showing (1) the continental shelf off the South Shetland Islands Archipelago at the Drake Passage, (2) the Bransfield Strait, (3) the Gerlache Strait, (4) the Antarctic Peninsula, (5) the Filchner Trough, (6) Cape Norvegia, and (7) the Austasen region. Each bathymetric isoline represents 500 m water depth. Map developed with QGIS software v. 3.8 and the International Bathymetric Chart of the Southern Ocean (IBCSO) shape files [44]. (**b**) Organic carbon content (weight %) in continental shelf sediment below the upper mixed layer (below the layer where physical and biological disturbance is more intense). The upper mixed layer thickness varied between 0 and 10 cm for the different stations. AUS, FTE, FTX, FTS, FTW, LAR, NWW, BFS, and DPS stand for Austasen, Filchner Trough East, Axis (X), Slope (S), West (W), Larsen, Northwestern Weddell Sea, Bransfield Strait, and Drake Passage, respectively. Map modified after Isla, 2020 [45]. Yellow dots show sampling stations.

On the one hand, it has been observed that at the beginning of the autumn, just after the summer bloom, the total lipid concentration in the sediment (e.g., labile organic material) is significantly higher than that of carbohydrates (e.g., refractory organic matter), which in turn, are more abundant at the beginning of the spring when the summer organic energy surplus has been exhausted and the system awakens after the dark season [43,46]. The seasonal pattern in sediment chemistry demonstrates the fact that benthos efficiently profits the pelagic input of fresh organic matter. Further support comes from the similar organic ^14^C signals (e.g., representing the most modern organic material) found in phytoplankton and holothurian tissue, which also suggests a selective organic matter ingestion by macrobenthic deposit feeders [47,48].

On the other hand, observations made a few meters above the sea floor showed that the magnitudes of water current velocity and sediment flux have inverse relationships, and tides are efficient water currents that distribute sediment over the shelf [49]. The incessant tidal force develops a settling–resuspension sequence that keeps organic matter available in the water column and traveling along the shelf, thus distributing energy over space and enabling autumn and winter macrobenthic trophic activity, also in regions of the shelf where the wind and summer primary production couplings were not generous. Underwater images showed that the abundances of deposit feeders and phytodetritus (e.g., green mats) coincided on broad shelves, supporting the hypothesis that deposit feeders profit most from intense sedimentation events, perhaps in combination with low current velocity [38]. Contrastingly, in faster water current environments on narrow shelves, suspension feeders were more abundant, suggesting a better adaptation to the lateral transport of organic particles suspended in the water column.

The highly diverse macrobenthic communities inhabiting the eastern WS continental shelf are mainly constituted by sessile suspension feeders, which provide a structure where many species thrive [50,51]. These assemblages can develop biomasses > 14,500 g m^2^ (wet weight) mainly due to the presence of massive hexactinellid “glass” sponges that can grow large (>1 m height) silicon structures [52]. One of the factors stimulating the high biodiversity in these communities is devastation caused by iceberg scouring, which is among the most significant disturbances for any large ecosystem [53]. In general, to the east of the WS, off Atka Bay, macrobenthic communities structured by suspension feeders are the most abundant, whereas to the west, deposit feeder taxa become more numerous and even dominate to the west of the Filchner Trough and off the northern AP, where the sections A and B of the Larsen Ice Shelf existed few decades ago [50,54]. There are also regional differences in macrobenthic biomass, the largest biomasses (>2000 g m^−2^) are associated with suspension feeder communities (e.g., off Austasen) and the smallest are associated with deposit-feeding assemblages (e.g., off the eastern AP) [22,55]. However, within the observed regional patterns, there is a patchy distribution of macrobenthic communities along the WS continental shelf that still lacks a conclusive explanation [38]. Several factors may influence the observed patchiness, from geophysical features such as the width of the continental shelf and water current velocity (e.g., modified after iceberg groundings) to an ongoing adaptation to the periodic and intense pelagic energy fluxes (e.g., plankton debris) developing since the last glacial maximum, modulated by processes such as iceberg scouring, biological competence (e.g., high abundances of deposit feeders can cause an elevated mortality of juvenile suspension feeders), reproductive strategy (e.g., budding and larval releases), and diversity in lithogenic sediment inputs [38,50,56,57].

Off the AP, at the northwestern WS, the BS, and the DP continental shelves, macrobenthic biomasses varied between 70 g m^−2^ and 530 g m^−2^ [58,59]. These values showed that there is no significant latitudinal macrobenthic biomass cline from the eastern WS to the shelf off the SSI at the DP [58], in spite of the distinct climatic conditions occurring at the sea surface (e.g., sea ice spatial and temporal distributions, ice shelf cover, wind force, and primary production). However, a macrobenthic community distribution pattern was observed along the WS shelf, moving from filter-feeding to deposit-feeding assemblages from open waters to the east to more persistent sea ice-covered waters to the west [58,59], which were perhaps also modulated by the continental shelf width differences among regions (e.g., broader to the west) and the associated water current regime (e.g., weaker on broader shelves) [50].

## 5. Discussion

### 5.1. Changing Ocean

Considering the fact that the climatic conditions at the sea surface may not produce significant differences among the OC contents in the sediment column (e.g., remaining available energy) (Figure 1b) and macrobenthic biomass distributions [58], the environmental controls at the sea floor may counterbalance the observed outcome. The water current velocity near the seabed can benefit filter-feeding over deposit-feeding species but also efficiently redistributes mineral and biogenic matter (including larvae) over the WS continental shelf [60,61], favoring a homogeneous distribution of particles and energy (e.g., organic carbon) along the shelf. Further evidence comes from chemical and photographic observations in areas that are under extinct and floating ice shelves. Five years after the collapse of Larsen Ice Shelf sections A and B, the presence of bacterial and detrital-related fatty acids in continental shelf sediment collected under the area where the ice shelf existed indicated that there was an input of “old” refractory organic matter into the region before the ice shelf collapsed [62]. To the south, photographic records taken hundreds of kilometers away from the Filchner Ice Shelf front and >1200 m water depth showed a benthic community thriving far from the fresh input of the recently produced organic matter [63]. This community was apparently structured by sponges (e.g., filter-feeding animals), which benefited from the lateral transport of organic particles. It is unlikely that the general water current pattern in the region may change in the near future, unless large tabular icebergs grounded on the continental shelf block the usual flow [64,65]. Hence, water currents may continue distributing energy as they currently do, keeping macrobenthic communities as we observe them today.

Nevertheless, the wind pattern is changing. As westerlies intensify over the SO associated with atmospheric temperature increases, water temperatures at the sea surface increase, ice shelf stability weakens, the swell intensifies, and the sea ice extent and atmospheric CO_2_ sink decrease [3,4,66,67]. All these processes have extensions into the energy transport mechanisms to the seabed and, consequently, in the maintenance of macrobenthic biomass; however, identifying the consequences of these relationships remains challenging. For example, observations for the eastern WS continental shelf showed that the sea ice extent has been locally increasing, at least until 2014, and apparently impacting macrobenthic biomass abundance and community composition, shifting from biomass-rich filter-feeding communities to biomass-poor deposit-feeding assemblages [68]. However, sea ice started decreasing by 2016/2017, with record-breaking low extents and unknown trends and consequences [4,69,70]. Further, based on the known macrobenthic communities’ patchy distribution in the area and the small station number sampled for the WS study (two to nine) relative to the area under investigation (hundreds of km^2^), the observed biomass abundance and community composition relationship to sea ice extent needs corroboration.

Primary production will increase together with space and light availability at the sea surface (e.g., after ice shelf collapses and sea ice extent reduction) and iron fertilization derived from the transit of icebergs [6,71] and ice shelf melting at the surface and grounding lines [12,19,20]. Further, due to warming and iron releases, polynyas may persist longer than usual, hosting vigorous biological production and downward energy fluxes over the continental shelf [19,20,21,66]. On the western coast of the AP, warming and the reductions in sea ice temporal and spatial extent are already increasing primary production [7] and benefiting cryptophytes over diatoms in the phytoplankton community assemblages [72,73,74], whereas projections into the future also point to an overall increase in primary production in a warmer SO [5]. Changes in the phytoplankton community composition may have an impact on the zooplankton assemblage, given that more cryptophytes than diatoms in a warmer and more stable water column with a shallow upper mixed layer may favor salp over krill populations, decreasing the magnitude of energy transfer to upper trophic levels and the seabed [72,75].

The present association among continental shelf macrobenthic biomass and sediment OC content distributions is the result of a “clockwork” mechanism of physical–biological processes, presumably interacting since the last glacial maximum when most of the continental shelf started losing its ice shelf cover (12,000 to 8500 years before the present, depending on the site) [56,76]. Nevertheless, ice shelf advances and retreats have been occurring along the Holocene, and, in fact, they are still going on [77], maintaining the long-term (centennial/millennial) macrobenthic community dynamics [56] and the challenge of estimating the pace of the macrobenthic response to natural ice shelf fluctuations. Due to anthropogenic force, ice shelf dynamics (e.g., retreats) undergo an unprecedented pace [78], adding uncertainty to the assessment of macrobenthic responses to natural environmental variation. Recent anthropogenic ice shelf collapses opened space for pelagic primary production [6] and a subsequent flux of recently produced labile organic matter, such as highly energetic lipids [79], which fuels continental shelf macrobenthic recolonization [54]. Most likely, these processes will continue over the WS shelf in the near future due to global warming [80]. However, it is difficult to predict how colonizing patterns may evolve given the slow growth rates of Antarctic macrobenthos and the relatively rapid (perhaps unprecedented) but heterogeneous ocean-warming pace in the region [22,23,81]. 

Among the various projected changes (e.g., ice shelf collapses, increases in iceberg scouring, and reductions in sea ice spatial and temporal extent), acidification and warming of the water column will affect the entire Southern Ocean already this century [1,81,82]. Acidification may produce severe changes in the efficiency of photosynthesis and the biological carbon pump; for example, important krill recruitment habitats could be seriously affected due to compromised hatching success [83], whereas OC fluxes (e.g., energy) to the seabed may decrease due to acidification impacts on foraminifera (e.g., reduction in shell weight) and flagellates (e.g., reduction in motility) and their importance in the Antarctic summer bloom [84,85,86]. By the end of the 20th century, atmospheric and sea surface warming was extraordinary in the vicinity of the AP relative to global averages, and after a cooling period during the first decade of the present century, the AP region seems to be warming at a rapid and record-breaking pace again [28,87,88,89]. The current warming at the sea surface converges with an ongoing warming of the water column concentrated within the Antarctic Circumpolar Current at 700 m to 1100 m water depth [81]. The observed warming accounted for almost 0.2 °C during the period 1950–1980, which is faster than the global ocean average and comparable to the atmospheric warming over the SO. Warming of the deep water column is especially serious for macrobenthic communities thriving on the continental shelf because their animal components are mostly sessile and they have evolved in cold (<0 °C) and constant environmental conditions [90]. These conditions modulated a rich and diverse assemblage of stenothermal communities with long generation times (therefore, reduced opportunity to mutate), which may require hundreds of years for adaptation [23]. Deep water warming implies a grave fitness challenge because tidal-driven warm water intrusions onto the continental shelf already expose macrobenthic communities to periodical changes larger than 2 °C (absolute). These communities cannot escape or migrate to colder environments because they live at the southern border of the Southern Ocean, and their persistence depends on their reduced genetic plasticity to cope with environmental stress and allow survival until adequate genetic adaptations can be achieved [23,91]. Based on the macrobenthic community composition and the projected warming over the continental shelf of the WS, the communities currently thriving to the east of the WS (e.g., off Austasen and the Drescher Inlet) have been identified as the most vulnerable to heat increments because they will be exposed earlier than the rest to the projected warming scenario and host the largest fraction of sessile organisms [22,82]. These communities build up biomasses between 2 g C m^−2^ and 88 g C m^−2^ and their collapse has the potential to liberate up to 299,000 tons of C to the water column and eventually to the atmosphere; this was calculated only considering a section of approximately 3400 km^2^ of the WS shelf [22]. For the rest of the WS and the vicinity of the AP shelves, the balance may not be different (perhaps with smaller releases of C), and most likely, warming will affect their current macrobenthic seascape earlier than other factors. Particularly in the case of the eastern AP, where the macrobenthic communities thriving on a continental shelf were recently liberated from its ice shelf cover (early 21st century) and currently show early stages of colonization [54], may not have the chance to reach the mature stage observed for the abundant communities to the east [22]. On the western side of the AP, however, the synergy among disturbance agents (e.g., warming, acidification, and transit of icebergs) may be already generating a stronger impact on the benthic communities thriving there [1], and there is not a baseline to contrast it. Whereas for the shallow coastal zone on the eastern AP, global change impacts, such as massive sediment loads derived from glacier retreats and iceberg scouring, have been documented [92,93] and even projected as a powerful negative feedback on climate change (e.g., blue carbon development) [93], the impact of global change on the Antarctic continental shelf benthos is mostly unknown. Nevertheless, general estimations have suggested a figure of blue carbon growth > 100 × 10^6^ t C y^−1^, which equals a value of up to USD 21.3 billion, showing the huge potential for carbon growth that the Antarctic continental shelf macrobenthic communities have and adding support to advocate for their protection [94,95].

### 5.2. Potential Future Developments

The analyses of the OC content in sediment suggest that temperature compromises the capacity of the sediment column to store carbon in the long term (e.g., centuries) [96]. In contrast to the similar magnitudes of primary production and chlorophyll-a abundance at the sea surface [10,29], the OC content in continental shelf sediment samples off the northern coast of the SSI in the DP was lower (by two-thirds) than that in continental shelf sediment off the western AP in the BS [97]. Given the comparable macrobenthic biomasses on the shelves of the SSI, also considering the highly heterogeneous environment in the AP vicinity [58,59], the water temperatures near the seabed, which are up to 4 °C higher at the DP than at the BS [30,31], emerged as the potential cause for the observed OC content differences [97]. Based on these observations, together with evidence acquired in the global ocean [96], it is likely that the ongoing ocean warming will reduce the amount of organic energy available in the sediment for continental shelf macrobenthic communities. Environmental projections forecast that primary production in the WS-AP region will increase, very probably the flux of energy to the seabed will accompany, together with warming; nevertheless, anticipating the way the benthic community will evolve in the future warmer ocean and estimating the amount of carbon retained in the sediment are some of the most challenging enterprises for Antarctic science. In this scenario, the actual macrobenthic communities, at least in the shape they currently show, appear as the losing parties in the future seascape because most likely, their species’ composition will be modified. Certainly, there will be winners: colonizers from other latitudes and other organisms, which may successfully adapt to ongoing environmental changes, may occupy the empty niche, benefiting from the energy surplus projected for the future ocean. Species with the ability to cope with warmer water may have more chances to persist, but experimental effort is still needed to identify the most fitting candidates for the actual Antarctic biodiversity pool.

The Antarctic macrobenthos is a big, valuable biodiversity library on fire that we have not yet had the time to read, and most likely, many of its books will be lost before we can open them. The establishment of marine protected areas (e.g., the Weddell Sea) is a necessary action if society attempts to maintain one of the valuable ecosystem services (e.g., biodiversity maintenance) that the Antarctic macrobenthos provides; however, it is only a first step if we acknowledge the danger that warming and acidification already represent for the rich and abundant Antarctic continental shelf macrobenthic life.

## 6. Conclusions

In the Southern Ocean, increases in physical energy (e.g., heat and wind force) enhance ice shelf disintegration and reduce sea ice extent. Consequently, primary production finds more space to bloom, increasing the amount of chemical energy (e.g., organic carbon) that can be exported to the deep sea. The macrobenthic fauna thriving on the continental shelf is expected to benefit from the predicted energy surplus; however, macrobenthic fauna is also exposed to anthropogenic warming. Most likely, in the near future, warming and acidification may modify the current continental shelf macrobenthic community compositions. This change, together with a reduced capacity of the sediment column to store carbon due to warming, makes the prediction of how the altered energy balance will shape the continental shelf macrobenthic communities very difficult. It is challenging to identify which species will better fit into a more energetic environment; however, it is viable that the current assemblages will change, leaving behind a valuable biodiversity pool. This loss, together with the important amount of blue carbon that the continental shelf macrobenthos represents, advocate for the protection of this unique marine fauna. The enterprise claims for rapid action, where the declaration of marine protected areas is an urgent first step given the pressure that warming and acidification pose over the old, abundant, and biodiverse Antarctic continental shelf fauna.

## Data Availability

All the data analyzed in the study come from the references listed along the text.

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
