# Peer review of "Animal–Energy Relationships in a Changing Ocean: The Case of Continental Shelf Macrobenthic Communities on the Weddell Sea and the Vicinity of the Antarctic Peninsula"

_biology, 2023, doi:10.3390/biology12050659_

Round 1

Reviewer 1 Report

The author has presented an interesting synthesis of many years of work focussing on the energy pathways studied in the Weddell Sea and Antarctic Peninsula region. He’s conclusion is that the outlook for biology (and specifically benthic biology) in the region is bleak due to a warming scenario and an assemblage with little room for adaptation. In summary he uses the analogy of a library on fire with many books being burned before they are able to be read and calls for the establishment of a Weddell Sea marine protected area as an urgent “first step” mitigation measure. The article is generally well written (although some checks should be done for sentence structure to ensure the correct interpretation, and a consistent use of tense, especially in the abstract) and contributes to a growing voice of literature calling for Antarctic marine protected areas for biodiversity and ecosystem service posterity.  I would suggest citing other similar calls for MPAs around the Antarctic, and consider the slightly different perspective brought by these articles: 

Political/Legal/Technical MPA perspective: Gogarty et al. 2019 https://doi.org/10.1080/14693062.2019.1694482

General Science perspective: Bax et al. 2021 https://onlinelibrary.wiley.com/doi/10.1111/gcb.15392

Simplified general perspective: Sands et al. 2023 https://tos.org/oceanography/article/perspective-the-growing-potential-of-antarctic-blue-carbon

I support the publication of this article with (very) minor changes. The most important is to check units of measurement used throughout with the published sources. The standout erroneous measure on line 135 on my copy “17 Tg C m-2 d-1 [31]”. This is supposed to be the amount of carbon currently (or in 1998) across the entire Weddell Sea Shelf (as opposed to per square meter per day). I imagine this amount of carbon squeezed into even a cubic meter would form a black hole.

I would like to throw in an alternative view that may be worth thinking about long term. It starts with the assumptions of the low adaptability of the Antarctic benthic fauna based on long stenothermic evolutionary history and long generation times. I think these assumptions require extra thought and here are some alternative views. 

The long evolutionary history with a substantial time period of isolation (18-34my) has shaped a unique benthic shelf fauna. During these times there have been very long (and repeated) periods of glaciation more severe than today where the entire shelf was bulldozed clear of benthos, requiring slope or deepsea refuges (Thatje et al 2005). These refuges are likely to be warmer (Antarctic deep water is 2-4°C) than the usual shelf water (0°C). As the cold glacial periods were longer than the brief interstadials, perhaps the fauna is adapted to warmer water after all.  

Adaptation is very different to phenotypic plasticity. If phenotypic plasticity exists (tends to be an adaptation to changing environments on an ecological scale) adaptation is not required. 

Adaptation/selection occurs over very short time frames (not thousands of years) and is strongly controlled by genetic diversity which is a function of population size (infinite population, infinite diversity) and mutation rate.  If a strong selection pressure is enforced on a very large population, the genetic diversity that allows the population to survive is already part of the population (even if hidden in very low frequency recessives) and over very few generations will be the dominant form of the diversity. There are some very good (and fairly simple) exploratory tools that demonstrate this (the R package learnPopGen for example).

The point I am making is that in my mind there is a very good chance that the benthic fauna is well adapted to warming and cooling events. Pulin (2002, TREE) suggests the balance between brooding and dispersing life histories is an example of the benthos ready for changes in shelf habitat loss. Clark and Crame (1992, Phil Trans Roy Soc) hypothesize that the cyclic nature of shelf habitat loss and gain over the glacial periods has promoted biological diversity – calling it the biological diversity pump.

Despite anthropogenic climate change, the global mean temperature is still less than previous interstadials. I do agree that assemblage composition changes are happening, and that there is an unprecedented threat of invasives given the amount of shipping for fishing, tourism, science and infrastructure support happening around the Antarctic, but the ability of the Antarctic shelf benthic fauna to adapt and evolve is not one that (in my mind) is a concern regarding biodiversity loss.

Something that has got me thinking is the “potential” mentioned in Bax et al. 2021 and Sands et al. 2023 listed above. Barnes (2015 Current Biology) demonstrates how sea-ice cover is directly related to growth in bryozoans, in that seas (or shelf areas) with different sea-ice cover show different rates of growth. The Weddell Sea showed the lowest growth (or change in growth over time) of all the Antarctic shelf seas as it had the most consistent sea-ice cover, where as other seas where sea-ice has been documented as having less sea-ice days, bryozoan growth has been increasing over time. It would be interesting to go back to the Weddell (and other) Sea(s) to re-sample the bryozoans to add a new time point since the trend of sea-ice reduction has become Antarctic wide (post 2014). It may be that the Weddell Sea has a huge potential for increased productivity – an alternative reason for protection.

Specific points to address:

Please check the summary and Abstract to ensure consistent use of tense

Final line of abstract may need tweaking as it doesn’t quite make sense, perhaps “The Antarctic macrobenthos biodiversity pool is a valuable ecosystem service that is under serious threat. Establishing marine protected areas may not be sufficient to preserve it.”

Line 86 “The deep continental shelf contributes making near-bottom…” contributes what?

Line 91-94 I’ve highlighted as it appears to confuse adaptation and phenotypic plasticity. I also remember a paper I think by Ducklow Clarke and Merrideth (2009ish, although I cant find it so may be wrong) measuring temperatures as warm as 4°c washing over the shelf in regions such as west Antarctic Peninsula and Amundsen Sea shelf.

Line 132 “When seasonal sea ice diminish, …” suggest “diminishes”

Line 135-136 Cant possibly be 17 Tg C m-2 d-1. The cited paper suggests this is the carbon content of the sediments of the entire Weddell Sea shelf. Might be worth checking some of the other cited values for correct units.

Line 158-159 it would be nice to see a reference to primary literature describing the location of the upper mixed layer in Antarctic sediments (although I haven’t found one). Perhaps an oxygenation test through the core?

Line 205 “periodic an intense” I guess should be “periodic and intense”.

Line 272 The last glacial maximum here is stated as 12000 to 8500 ybp. Perhaps there is an issue with what is meant by last glacial maximum – in my mind this is the coldest period of a glaciation event, which I think was 18,000 ybp, whereas 12,000 – 8,500 was the transition between the glacial to the present interstadial.

Line 290 don’t forget to replace XX century for…? 21st?

The paragraph beginning “Among the various projected changes…”, line 282, I have various notes relating to the inaccurate interpretation of how adaptation works and the difference between adaptation and phenotypic plasticity (not the same). See discussion above.

Line 313 I think the figures quoted here need a temporal element to the units:  2 g C m2 [per day?]

Line 346 “future warmer ocean is one the most challenging” requires an “of”: “future warmer ocean is one OF the most challenging”

Author Response

Reviewer 1

The author has presented an interesting synthesis of many years of work focussing on the energy pathways studied in the Weddell Sea and Antarctic Peninsula region. He’s conclusion is that the outlook for biology (and specifically benthic biology) in the region is bleak due to a warming scenario and an assemblage with little room for adaptation. In summary he uses the analogy of a library on fire with many books being burned before they are able to be read and calls for the establishment of a Weddell Sea marine protected area as an urgent “first step” mitigation measure. The article is generally well written (although some checks should be done for sentence structure to ensure the correct interpretation, and a consistent use of tense, especially in the abstract) and contributes to a growing voice of literature calling for Antarctic marine protected areas for biodiversity and ecosystem service posterity. 

I thank Reviewer 1’s for the constructive and didactic comments provided in his/her revision.

I would suggest citing other similar calls for MPAs around the Antarctic, and consider the slightly different perspective brought by these articles: 

Political/Legal/Technical MPA perspective: Gogarty et al. 2019 https://doi.org/10.1080/14693062.2019.1694482

General Science perspective: Bax et al. 2021 https://onlinelibrary.wiley.com/doi/10.1111/gcb.15392

Simplified general perspective: Sands et al. 2023 https://tos.org/oceanography/article/perspective-the-growing-potential-of-antarctic-blue-carbon.

Based on the suggested studies of Bax et al., 2021 and Sands et al., 2023, I included in the Discussion and the Conclusions sections, in lines 358 to 361 and 409 to 410, respectively, the suggestion that the blue carbon value that macrobenthic communities represent, provides further support to advocate for the protection of Antarctic continental shelf macrobenthos.

I support the publication of this article with (very) minor changes.

Thanks for the support.

The most important is to check units of measurement used throughout with the published sources. The standout erroneous measure on line 135 on my copy “17 Tg C m-2 d-1 [31]”. This is supposed to be the amount of carbon currently (or in 1998) across the entire Weddell Sea Shelf (as opposed to per square meter per day). I imagine this amount of carbon squeezed into even a cubic meter would form a black hole.

Thanks for the observation, it was typo, the correct units are Tg C y-1 (not square meter per day) and correspond for the entire Weddell Sea continental shelf budget. The units are corrected in the revised version in line 156.

I would like to throw in an alternative view that may be worth thinking about long term. It starts with the assumptions of the low adaptability of the Antarctic benthic fauna based on long stenothermic evolutionary history and long generation times. I think these assumptions require extra thought and here are some alternative views. 

The long evolutionary history with a substantial time period of isolation (18-34my) has shaped a unique benthic shelf fauna. During these times there have been very long (and repeated) periods of glaciation more severe than today where the entire shelf was bulldozed clear of benthos, requiring slope or deepsea refuges (Thatje et al 2005). These refuges are likely to be warmer (Antarctic deep water is 2-4°C) than the usual shelf water (0°C). As the cold glacial periods were longer than the brief interstadials, perhaps the fauna is adapted to warmer water after all.  

Adaptation is very different to phenotypic plasticity. If phenotypic plasticity exists (tends to be an adaptation to changing environments on an ecological scale) adaptation is not required. 

Adaptation/selection occurs over very short time frames (not thousands of years) and is strongly controlled by genetic diversity which is a function of population size (infinite population, infinite diversity) and mutation rate.  If a strong selection pressure is enforced on a very large population, the genetic diversity that allows the population to survive is already part of the population (even if hidden in very low frequency recessives) and over very few generations will be the dominant form of the diversity. There are some very good (and fairly simple) exploratory tools that demonstrate this (the R package learnPopGen for example).

The point I am making is that in my mind there is a very good chance that the benthic fauna is well adapted to warming and cooling events. Pulin (2002, TREE) suggests the balance between brooding and dispersing life histories is an example of the benthos ready for changes in shelf habitat loss. Clark and Crame (1992, Phil Trans Roy Soc) hypothesize that the cyclic nature of shelf habitat loss and gain over the glacial periods has promoted biological diversity – calling it the biological diversity pump.

Despite anthropogenic climate change, the global mean temperature is still less than previous interstadials. I do agree that assemblage composition changes are happening, and that there is an unprecedented threat of invasives given the amount of shipping for fishing, tourism, science and infrastructure support happening around the Antarctic, but the ability of the Antarctic shelf benthic fauna to adapt and evolve is not one that (in my mind) is a concern regarding biodiversity loss.

Something that has got me thinking is the “potential” mentioned in Bax et al. 2021 and Sands et al. 2023 listed above. Barnes (2015 Current Biology) demonstrates how sea-ice cover is directly related to growth in bryozoans, in that seas (or shelf areas) with different sea-ice cover show different rates of growth. The Weddell Sea showed the lowest growth (or change in growth over time) of all the Antarctic shelf seas as it had the most consistent sea-ice cover, where as other seas where sea-ice has been documented as having less sea-ice days, bryozoan growth has been increasing over time. It would be interesting to go back to the Weddell (and other) Sea(s) to re-sample the bryozoans to add a new time point since the trend of sea-ice reduction has become Antarctic wide (post 2014). It may be that the Weddell Sea has a huge potential for increased productivity – an alternative reason for protection.

Many thanks for the sound, thoughtful and didactic alternative view to contrast the idea of the low adaptation ability to the ongoing increasing temperature trend of the Antarctic benthic fauna, due to its long stenothermic evolutionary history and long generation times. I liked very much the idea that, perhaps the benthic fauna is adapted to warmer water after all and that the genetic diversity required to survive, is already there in the population. While I share Reviewer 1’s thoughts and, in my opinion, the proposed alternative opens a research avenue; I also acknowledge that, the experimental effort to assess the benthic response to ongoing warming is challenging due to the high diversity of Antarctic macrobenthos and the logistics it requires to collect and place the different specimens in good conditions in the laboratory. To avoid potential misunderstandings, I changed the term “adaptation” in lines 303 and 307 in the original manuscript for “fitness” in line 303 (line 331 in the revised version) and “cope” in several lines along the revised version. Given that Reviewer 1, among other authors (e.g., Griffiths et al., 2017; Peck 2018), shares the idea that the current Antarctic macrobenthic community assemblages will be modified due to ongoing global change and that I am not an expert in adaptation abilities, I prefer to treasure the proposed alternative for potential future contributions and collaborations, where these concepts and ideas can find the dedication to their development that they deserve. On the lack of experimental evidence for growth rates in the various species and a solid description of the potential future macrobenthic community composition (e.g., a switch from sponge dominated communities to bryozoan dominated assemblages), I would like not to mention the potential the Weddell Sea has for future increased productivity as an additional reason for protection. However, I acknowledge the huge blue carbon value it already has and this is included in the revised version of the manuscript, based on Reviewer 1’s suggestions.

Specific points to address:

Please check the summary and Abstract to ensure consistent use of tense.

I’ve checked and corrected both sections; hopefully, the tense issues are fulfilled. An American native English speaker helped in revision of the English language and improved the English syntax and grammar of the original version.

Final line of abstract may need tweaking as it doesn’t quite make sense, perhaps “The Antarctic macrobenthos biodiversity pool is a valuable ecosystem service that is under serious threat. Establishing marine protected areas may not be sufficient to preserve it.”

Suggestion accepted.

Line 86 “The deep continental shelf contributes making near-bottom…” contributes what?

The sentence “The deep continental shelf contributes making near-bottom current circulation sluggish, …” has been changed for “The deep continental shelf contributes making near-bottom water current circulation sluggish, …” in line 106 in the revised version.

Line 91-94 I’ve highlighted as it appears to confuse adaptation and phenotypic plasticity. I also remember a paper I think by Ducklow Clarke and Merrideth (2009ish, although I can’t find it so may be wrong) measuring temperatures as warm as 4°c washing over the shelf in regions such as west Antarctic Peninsula and Amundsen Sea shelf.

I maintained the term “cope” to avoid confusion with adaptation (evolutionary).

Line 132 “When seasonal sea ice diminish, …” suggest “diminishes”

Suggestion accepted.

Line 135-136 Cant possibly be 17 Tg C m-2 d-1. The cited paper suggests this is the carbon content of the sediments of the entire Weddell Sea shelf. Might be worth checking some of the other cited values for correct units.

The units in the original version were replaced for Tg C y-1

Line 158-159 it would be nice to see a reference to primary literature describing the location of the upper mixed layer in Antarctic sediments (although I haven’t found one). Perhaps an oxygenation test through the core?

Thanks for the comment. I understand the need for a clearer description. The horizon “upper mixed layer” of the sediment column is commonly described as the layer, where the distribution pattern of any parameter is rather uniform. In Figure1, the upper mixed layer was defined using 210Pb activity and organic carbon concentration signals (e.g., Isla et al., 2004, Isla, 2016; Isla and DeMaster, 2018, 2021). The distribution pattern of the different parameters changed from core to core (location to location) mainly due to the local water current intensities (sediment resuspension events) and benthic activity (bioturbation). Based on my own experience, it is not possible to establish a general upper mixed layer depth even for a single region. Perhaps, this is the reason why it is difficult to find such value in the primary literature. The organic carbon concentration values provided in Figure 1 were obtained for the fraction of the sediment cores below the upper mixed layer on individual basis. Figure 1 caption changed to “Organic carbon content (weight %) in continental shelf sediment below the upper mixed layer (below the layer where physical and biological disturbance is more intense). The upper mixed layer thickness varied between 0 and 10 cm for the different stations.”

Line 205 “periodic an intense” I guess should be “periodic and intense”.

Correct!

Line 272 The last glacial maximum here is stated as 12000 to 8500 ybp. Perhaps there is an issue with what is meant by last glacial maximum – in my mind this is the coldest period of a glaciation event, which I think was 18,000 ybp, whereas 12,000 – 8,500 was the transition between the glacial to the present interstadial.

In the manuscript, I meant the time interval from 12000 to 8500 ybp as the time when most of the Antarctic Peninsula-Weddell Sea continental shelf region lost most of its ice shelf cover, which varied from region to region according to the time intervals provided in Hall (2009) and Gutt & Koltun (1995) (Hall, B. Holocene glacial history of Antarctica and the sub-Antarctic islands. Quat. Sci. Rev. 2009, 28, 2213–2230; Gutt, J.; Koltun, V.M. Sponges of the Lazarev and Weddell Sea, Antarctica: explanations for their patchy occurrence. Ant. Sci. 1995, 7, 227-234.). To avoid misunderstandings, I changed the sentence for “started losing” instead of “lost” and simply moved the time interval to end of the sentence as “… the last glacial maximum, when most of the continental shelf started losing lost its ice-shelf cover (12000 to 8500 years before present, depending on site)…” in lines 295 to 297 of the revised version.

Line 290 don’t forget to replace XX century for…? 21st?

Thanks, XXth replaced for 20th.

The paragraph beginning “Among the various projected changes…”, line 282, I have various notes relating to the inaccurate interpretation of how adaptation works and the difference between adaptation and phenotypic plasticity (not the same). See discussion above.

Thanks again for the constructive comments; as explained above in the response to previous comments, I changed the term “adaptation” in the revised version of the manuscript.

Line 313 I think the figures quoted here need a temporal element to the units:  2 g C m2 [per day?].

The figure represents the amount of C measured in the samples of macrobenthic communities collected with multibox corers (Gerdes D, 1990. Antarctic trials of the multi-box corer, a new device for benthos sampling. Polar Record 26, 35-38.). It represents the amount of C in the organisms; thus, it does not require the temporal element in the units.

Line 346 “future warmer ocean is one the most challenging” requires an “of”: “future warmer ocean is one OF the most challenging”

Again, it is correct!

Reviewer 2 Report

The paper tries to perform a compendium of the scientific knowledge about the Antarctic peninsula both from chemical and physical points of view and their relations with macrobenthic aspects. For this the study can be counted as a review paper even if some things should be revised.

In particular, a part related to the history of the Antarctic peninsula studies should be added. Furthermore, many references appear to have been entered as self-references. Of course that the author is an Antarctic peninsula expert but within the paper there are 19 works signed by the author and some of them are not very pertinent to the text.

For example, in line 85 the reference [16] does not seem very relevant. The same thing goes for the reference [26] in line 121 which seems redundant since it has been cited within the reference [7] here cited again. The reference [27] in line 119 is redundant since it is present within [7]. From lines 124 to 130 the same reference [27] is repeated twice and it is not clear why since they are two consecutive periods.

These patterns are present throughout all the paper (see for example reference [34], [37], [46], etc…) making the review more like a summary of the investigations written by the author than a real examination and comparison of all the results obtained about Antarctic peninsula. I would therefore advise the author to consider a complete review of the work by referencing to other studies than his own (which cover more than 20%) and to re-propose the paper whose usefulness would be undoubted.

Given the need for a review article on this topic I do not reject the article but ask for major revision.

Author Response

Reviewer 2

The paper tries to perform a compendium of the scientific knowledge about the Antarctic peninsula both from chemical and physical points of view and their relations with macrobenthic aspects. For this the study can be counted as a review paper even if some things should be revised.

In particular, a part related to the history of the Antarctic peninsula studies should be added. Furthermore, many references appear to have been entered as self-references. Of course that the author is an Antarctic peninsula expert but within the paper there are 19 works signed by the author and some of them are not very pertinent to the text.

I thank Reviewer 2 for her/his comments and observations. I acknowledge the manuscript refers to several previous studies of the author; however, the reason to include them in the manuscript is to provide support for the various ideas and figures provided in the present work and in many cases, in absence of articles from other authors, supporting the ideas developed in the present work. It is not the purpose to self-reference, it is to provide solid evidence, show that the various processes described in the manuscript take place along the continental shelves of the Weddell Sea and the Antarctic Peninsula and not only in few particular spots. Given that I cannot justify why I cited my works every time I did in the response letter (it will be represent another manuscript), as an example, I will do it in the examples Reviewer 2 provided. Nevertheless, I revised the information related to the western coast of the Antarctic Peninsula and in the revised version, my works are only 4 vs. 23 from other authors. I referred to the studies which I collaborated with, only when the idea in the sentence needed support. For the studies on the eastern coast of the Antarctic Peninsula, the proportion of studies with my collaboration is larger, simply because the groups I have been working with, have produced most of the information needed to assess ongoing climate change effects on the animal-energy relationships in that region; therefore I cannot take them away.

For example, in line 85 the reference [16] does not seem very relevant.

Yes, from the title, reference 16 (Isla, E.; Gerdes, D. Ongoing ocean warming threatens the rich and diverse macrobenthic communities of the Antarctic continental shelf. Prog. Oceanogr. 2019, 178, 102180, doi.org/10.1016/j.pocean.2019.102180.), does not seem relevant. However, this study provides evidence for tidal currents occurring near the seabed (e.g., > 300 m water depth) along >3000 km of the Weddell Sea continental shelf, I referred to this study just to let clear that these currents develop all along the Weddell Sea continental shelf. On the other hand, reference 15 (Isla, E.; Gerdes, D.; Palanques, A.; Gili, J-M.; Arntz, W. Particle fluxes and tides near the continental ice edge on the eastern Weddell Sea shelf. Deep-Sea Res. II 2006, 53, 866-874.), provides evidence for the relationship among tidal currents and sediment transport, which together with heat transport, is among the facts I would like to highlight for the processes taking place on the region of the Antarctic continental shelf related to the present work. In any case, I removed reference 15 from the sentence and let reference 16 standing alone, given that the reader may find evidence in reference 16 for the wide spatial coverage of the processes in disadvantage of the description of other processes such as sediment transport.

The same thing goes for the reference [26] in line 121 which seems redundant since it has been cited within the reference [7] here cited again.

I could not find reference 26 in line 121, it seems that Reviewer 2 referred to line 116. If this is the case, regarding my authorship in references 7 (Isla, E.; DeMaster D.J. Biogenic matter content in marine sediments in the vicinity of the Antarctic Peninsula: recent sedimentary conditions under a diverse environment of production, transport, selective preservation and accumulation. Geochim. Cosmochim. Acta 2021, doi.org/10.1016/j.gca.2021.04.021) and 26 (Masqué, P.; Isla, E.; Sánchez-Cabeza, J.A.; Palanques, A.; Bruach, J.M.; Puig, P.; Guillén, J. Sediment accumulation rates and carbon fluxes to bottom sediments at the Western Bransfield Strait [Antarctica]. Deep-Sea Res. II, 2002, 49, 921-933), both articles represent different studies developed in the vicinity of the Antarctic Peninsula and these were based on different station grids. One article may cite the other but the reason to cite both of them in the present work, is because their station grids together, provide evidence of processes occurring on regional rather than local settings.

The reference [27] in line 119 is redundant since it is present within [7].

Again, reference 27 was cited because it includes information from the almost 200 km long Gerlache Strait, located to the South of Bransfield Strait, showing that the sediment accumulation rates mentioned in the sentence in line 119 represent a longer area off the western Antarctic Peninsula than the Bransfield Strait alone. This information makes the comparison with the eastern Antarctic Peninsula values more solid, among other aspects, because of the similar spatial cover provided with both studies.

From lines 124 to 130 the same reference [27] is repeated twice and it is not clear why since they are two consecutive periods.

Thanks for the observation, the leading to reference 17 has been removed from the first sentence.

These patterns are present throughout all the paper (see for example reference [34], [37], [46], etc…) making the review more like a summary of the investigations written by the author than a real examination and comparison of all the results obtained about Antarctic peninsula.

The articles 34 (Isla, E.; Rossi, S.; Palanques, A.; Gili, J-M.; Gerdes, D.; Arntz, W. Biochemical composition of marine sediment from the eastern Weddell Sea [Antarctica]: high nutritive value in a high benthic-biomass environment. J. Mar. Sys. 2006, 360, 255-267.), 37 (Gerdes, D.; Isla, E.; Knust, R.; Mintenbeck, K.; Rossi, S. Response of benthic communities to disturbance: the artificial disturbance experiment BENDEX on the eastern Weddell Sea Shelf, Antarctica. Polar Biol. 2008, 31, 1469-1480.) and 46 (Sañé, E.; Isla, E.; Gerdes, D.; Montiel, A.; Gili, J-M. Benthic macrofauna assemblages and biochemical properties of sediments in two Antarctic regions differently affected by climate change. Cont. Shelf Res. 2012, 35, 53-63.), provide information from different scientific approaches and study areas, covering regions to the East (e.g., reference 46) and the West of the Weddell Sea (e.g., 34 and 37). One of the scientific problems in the Antarctic, is the rather small spatial cover that the research efforts provide relative to the huge extent of its marine environment; therefore, with this apparent unnecessary citing effort, I tried to provide the reader with as much evidence as possible for the various processes, representing the widest spatial cover possible.

I would therefore advise the author to consider a complete review of the work by referencing to other studies than his own (which cover more than 20%) and to re-propose the paper whose usefulness would be undoubted.

Given the need for a review article on this topic I do not reject the article but ask for major revision.

I rewrote some parts of the manuscript and added a Conclusions section, keeping the main ideas, just keeping the apparent unnecessary self-citation always in absence of replacing works; hopefully, Reviewer 2 finds the revised version of the manuscript better than the original version and suitable for publication.

Reviewer 3 Report

The author has addressed an interesting issue in the study area. 

I would suggest the author to re-organize and resubmit the MS. Major concerns are:

1. I failed to find the objectives of the study

2. I could not find the methods of the study - even for any review there should be method.

3. Please arrange the main findings on the basis of objectives or research questions.

4. Then add the discussion and recommendations

Author Response

Reviewer 3

The author has addressed an interesting issue in the study area. 

Thanks for the comment.

I would suggest the author to re-organize and resubmit the MS. Major concerns are:

  1. I failed to find the objectives of the study

Thanks again for this comment. Reviewer 3 did not fail; the objectives were not clearly stated. In the revised version objectives are provided in lines 58 to 61. “…develop a general picture on animal-energy relationships at the Weddell Sea and the western Antarctic Peninsula continental shelves, bringing together several physical, chemical, geological and biological aspects of the local environments and the macrobenthic fauna thriving there…”

  1. I could not find the methods of the study - even for any review there should be method.

There was no methods section in the original version of the manuscript, I based the present study on a literature review and own experience, as stated on lines 57 to 58 of the revised version.

  1. Please arrange the main findings on the basis of objectives or research questions.

I attempted to present the main findings along the discussion, where I describe the various changes in the environmental conditions and macrobenthic community abundance and composition, together with their potential developing trends and a suggestion on the regions that may experience changes earlier than in the rest of the study area.

  1. Then add the discussion and recommendations.

The information has been added in the revised version of the manuscript, where I included comments of Reviewers 1 and 2, and now is hopefully clearer for Reviewer 3 and finds it suitable for publication. The revised version includes a separate Conclusions section.

Round 2

Reviewer 3 Report

The authors have done some corrections. 
I thank them.

However, I would like see the clear objectives and the study methods for the study. 
Even for the review paper we can not just say “ based on literature and my experience’.

Please mention the sources of information and data. 
In the manuscript, I see Introduction and Discussion only. 
I would suggest to go through any good review paper and follow that.

Author Response

I thank Reviewer 3 for the time and dedication invested in a second revision of the manuscript.

The authors have done some corrections. 
I thank them.

The attitude is reciprocal.

However, I would like see the clear objectives and the study methods for the study. 
Even for the review paper we can not just say “ based on literature and my experience’.

I have included separate sections on Objectives and Methods to make clear the information. In lines 241 to 250 of the 2nd revised version of the manuscript it reads as follows:

2. Objectives

To provide a synthetic picture on the ongoing animal-energy relationships at the Weddell Sea and the western Antarctic Peninsula continental shelves and their potential consequences determined by the ongoing global change pressure on an apparently remote region of the ocean.

3. Methods

The synthetic information presented in this analysis follows a path from the sea surface to the seafloor using organic carbon budgets to structure the energy flow from primary production to macro-benthic biomass stocks, modulated by the current environmental controls at the sea surface and along the water column.

Please mention the sources of information and data. 

All the sources of information and data are linked to studies, which are listed in the References section.

In the manuscript, I see Introduction and Discussion only. 
I would suggest to go through any good review paper and follow that.

I hope the current revised version of the manuscript meets Reviewer 3’s suggestions and requirements.